# Relationship between Depression and Cognitive Inhibition in Men with Heroin or Methamphetamine Use Disorder in First-Time Mandatory Detoxification

**DOI:** 10.3390/healthcare11010070

**Published:** 2022-12-26

**Authors:** Yong Xin, Xiao Zhong, Xiaoqian Zhang, Youping Chen, Wei Xin, Chuanjun Liu, Haojie Fu, Chun Feng, Guoguo Zuo

**Affiliations:** 1Department of Psychology, Faculty of Law, Southwest University of Science and Technology, Mianyang 621010, China; 2School of Psychology, Beijing Sport University, Beijing 100084, China; 3Xinhua Drug Rehabilitation Center, Mianyang 610000, China; 4Sichuan Preschool Educators College, Mianyang 610000, China; 5Department of Sociology and Psychology, Institute of Psychology, School of Public Administrations, Sichuan University, Chengdu 610065, China; 6Shanghai Research Institute for Intelligent Autonomous Systems, Tongji University, Shanghai 200092, China

**Keywords:** compulsory drug treatment, addiction, depression, inhibition, substance abuse

## Abstract

Previous studies have shown that males with substance use disorder (SUD) in their first mandatory detoxification experience high rates of depression. It is unknown whether this high depression incidence contributes to impaired inhibition. In this work, two studies were undertaken to examine the role of depression in cognitive inhibition in heroin and methamphetamine withdrawal. We used the Beck Depression Inventory (BDI) and the self-control scale (SCS) to explore the relationship between depression and impulse inhibition in patients participating in mandatory drug treatment for the first time (Study 1). The results showed that depression negatively predicted impulse inhibition. The Stroop color–word interference task was used to explore the role of patients’ depression in their inhibitory abilities (Study 2). The results showed that the high-depression group had weaker inhibition performances in the Stroop color–word interference task compared to the low-depression group. This study shows that cognitive inhibition is weaker in people with high-depression addiction than in those with low depression. This result suggests that attention should be paid to the role of depressive comorbidity when conducting working memory training treatment for substance addiction.

## 1. Introduction

The World Drug Report 2022 shows that an estimated 284 million people (5.6% of the population) had used a drug in the past 12 months in 2020, and that relapse rates and the risk of relapse have increased in some countries during the pandemic [1]. The difficult withdrawal characteristics and high drug addiction relapse rates make drug issues a major problem for global health systems. Drug addiction is a disorder involving impulsive and compulsive factors, manifested by a marked deficit in compulsive drug-seeking and inhibitory control [2]. Inhibitory function or cognitive inhibition refers to the ability to effectively inhibit irrelevant task information or inappropriate behavior and is one of the main components of an individual’s executive function [3]. Effective cognitive inhibition protects task-relevant content in working memory from irrelevant information and prevents limited working memory capacity from being occupied by irrelevant information [4], such as reducing drug-seeking impulses. Cognitive inhibition is important for people with SUD to successfully withdraw from substances and prevent relapse after withdrawal [5].

The dual-systems model of substance addiction suggests that the behavioral processes of drug-seeking and eventual addiction stem from the interaction of neurocognitive systems, i.e., impulsive (reflex/automatic/spontaneous) processes and inhibitory (reflex/slower/negotiated) processes, both of which are determinants of behavior [6]. In terms of individual behavioral responses, attractive objects (e.g., drugs) refer to items that individuals have a strong impulse to respond to. For a goal to be accomplished (e.g., withdrawal), individuals need to quickly inhibit their attentional tendency to ease withdrawal from drugs and physical and psychological (emotional) responses and subsequently remain directed and focused on the goal [7,8]. The dual-process model suggests that the impulsive system should have a greater effect on behavior under conditions where the inhibitory system is weak or impaired, while the impulse should have a lesser effect on behavior under conditions where the inhibitory system is strong or intact [6,9]. Thus, the inhibitory system is seen as a “brake” on the impulsive system [6], helping to reduce the people with SUD’s impulse to seek out drugs and successfully abstain from them. Cognitive inhibition is an important mediator of drug addiction and may be an important target for treating people with SUD by improving impaired cognitive abilities [10,11].

However, previous studies have found severe impairment of inhibitory function in people with SUD [12,13,14]. For example, Farhadian et al. [15] found that drug addicts performed worse in the Stroop task. According to the stage model of addiction, the inhibitory function is involved in the behavioral process of pre-drug-seeking and eventual addiction [16]. Moreover, impairment of inhibitory function can lead to an imbalance in the addict’s impulse and inhibition systems, making it difficult for the individual to suppress strong impulses to use drugs [17], leading to the failure of the addict to quit drugs [14].

In the case of drug addicts, long-term substance abuse is a major contributor to their impaired cognitive functioning [13,18,19], and the impact effect varies with the type of drug. For example, methamphetamine users have longer response times in the Stroop task compared to opioid users [20]. In addition, some researchers are focusing on the role of depression in substance use, suggesting that depression has the effect of increasing the probability of substance use (i.e., increasing the likelihood of initiating use, promoting earlier emergence, and/or influencing the amount or persistence of use) [21]. For example, research has found that depression may lead to the initiation of or increase in the frequency of cannabis use [22]. Past research has found that worsening depressive symptoms can also impair an individual’s cognitive abilities [23], including attention, working memory, and cognitive inhibition [24].

Depressive symptoms have a negative role in drug withdrawal in addicts. Depressed individuals have difficulty removing negative information from working memory and exhibit negative self-information processing biases, attentional biases, and memory biases [25]. Moreover, according to a dual-process model of cognitive vulnerability to depression, highly depressed individuals engage in negative automatic processing that produces negative cognitive or emotional responses, which in turn diminishes cognitive resources and causes deficits in executive functions, such as inhibitory control [26], and as a result, the people with SUD fail to withdraw [14]. Therefore, it can be inferred that addiction with depression is associated with poorer inhibitory control than addiction without depression.

In addition, according to the Narcotics Control Law of the People’s Republic of China, the compulsory isolation drug rehabilitation system applies to those whose drug addiction is so serious that they are not suitable for community drug rehabilitation or refuse to accept community drug rehabilitation, take or inject drugs again during community drug rehabilitation, seriously violate the community drug rehabilitation agreement, or take or inject drugs again after community or compulsory isolation drug rehabilitation [27]. Patients participating in mandatory drug treatment for the first time belong to the first two categories, and they have not yet returned to normal life after receiving isolation and educational measures such as drug treatment management, psychological correction, physical rehabilitation, correction, and training, not to mention relapse. The chronic and recurrent nature of drug use can deepen the severity of cognitive deficits in patients [19], so patients participating in mandatory drug treatment for the first time may have a lower degree of substance use-related cognitive impairment and may be easier to treat. Due to the initial involvement in compulsory detoxification, this patient may be affected by a stressful lifestyle [28], and their negative emotions (e.g., depression) may be more intense.

In summary, previous studies have focused only on the association between persistent drug use and impairment in inhibitory functioning, without focusing on the role of factors such as depression in first-time participants in compulsory drug treatment. Therefore, focusing on patients participating in mandatory drug treatment for the first time, this work explored the role of depression in people with SUD in their inhibitory control through two studies based on the dual-systems model of substance addiction and the susceptibility dual-processing theory of depression, thereby informing comprehensive research and potential treatment targets in the cognitive domain related to substance use disorders.

## 2. Study 1: The Relationship between Depression and Impulse Inhibition

### 2.1. Methods

#### 2.1.1. Participants

A total of 571 males with SUD were randomly selected in a compulsory drug treatment management facility in Southwest China. Inclusion criteria were: (a) meeting the DSM-5 diagnostic classification criteria for substance abuse, (b) first-time participants in mandatory drug treatment, (c) age 18–60 years, and (d) agreement to participate in this project and signing an informed consent form. Exclusion criteria were: (a) severe physical illness, (b) education level below the sixth grade of elementary school, and (c) drug use in the previous 30 days. Finally, the mean age of the participants was 34.47 years (range = 18–58 years, SD = 7.360), and the mean age at first substance abuse was 24.33 years (range = 12–48 years, SD = 6.946). All these patients used methamphetamine or heroin as their primary drug of use before participating in mandatory detoxification.

#### 2.1.2. Study Instrument

Depression. The Beck Depression Self-Rating Inventory (BDI) was developed by Beck and is the most widely used scale in clinical psychology to assess depressive status in normal adults (18–60 years). This test uses a 13-item version of the BDI (such as “0 = I don’t feel depressed, 1 = I feel depressed or frustrated, 2 = I am depressed all day and can’t get rid of it, 3 = I am very depressed and can’t stand it anymore”), with higher total scale scores representing higher levels of depressive symptoms [29]. The total score of this scale ranges from 0 to 39, where 0−4: no clinically significant depression, 5−7: mild depressive symptoms, 8−15: moderate depressive symptoms, and >15: severe depressive symptoms [29]. In this study, Cronbach’s α coefficient for this scale was 0.818.Impulse inhibition. The impulse inhibition dimension of the short version of the self-control scale developed by Tangney et al. [30] was used, and this section contains six items, such as “I often act without thinking about it”. A five-point scale was used, and the higher the score, the better the impulse control. In this study, Cronbach’s α coefficient for this scale was 0.799.

#### 2.1.3. Data Analysis and Common Method Deviation Test

The quantitative data in this study were expressed as mean and standard deviation. All data were analyzed using IBM SPSS Statistics 25.0(IBM, USA; https://www.ibm.com/support/pages/downloading-ibm-spss-statistics-25) (accessed on 11 January 2022). A regression model with impulse inhibition as the outcome variable and depression as the predictor variable was developed to explore the relationship between depression and impulse inhibition. The test was α = 0.05.

The Harman one-way test was used to test for common method bias. The results revealed that four factors were generated without rotation, and the first factor had an explanatory rate of 24.158%, which was less than the critical criterion of 50% [31], indicating that there was no significant common method bias in this study.

### 2.2. Results

#### 2.2.1. Results of Descriptive Statistics and Correlation Analysis

The means, standard deviations, and correlation coefficients of depression and impulse inhibition are shown in Table 1. Correlation analysis showed that depression was significantly negatively associated with impulse inhibition (*p* < 0.001).

#### 2.2.2. Results of Regression Analysis

The results showed that the Durbin–Watson coefficient was 1.950, indicating that there was no significant autocorrelation among subjects in the sample. As shown in Table 2, depression positively predicted impulse inhibition (*b* = −0.031, 95% CI = (−0.043, −0.020).

### 2.3. Discussion of Study 1

Preliminary results from Study 1 suggest that cognitive inhibition in people with SUD differs significantly across levels of depression. Specifically, the group of highly depressed addiction patients showed significantly lower stimulation-induced inhibition than the low-depression group. This result is consistent with the research hypothesis that depression in addicts significantly and negatively predicts inhibition.

Patients who are participating in compulsory drug rehabilitation for the first time feel ill-treated because they are not accustomed to the compulsory drug rehabilitation environment, do not have confidence in their drug rehabilitation, are confused about their future, or believe that their addiction is not that significant, but they are in compulsory isolation for two years like everyone else [32,33]. Notably, symptoms of higher depression may cause individuals to engage in negative automatic processing, which weakens their cognitive resources [25] and, in turn, impairs their working memory capacity. Thus, individuals with high depression scores exhibit poorer impulse inhibition.

## 3. Study 2: The Role of Depression in Inhibitory Control

The results of Study 1 suggested that the group of highly depressed addicts had significantly lower cognitive inhibition than the low-depression group. To further determine the predictive relationship between the two, Study 2 used the Stroop task to examine the differences in inhibitory function between those with high and low depression scores, seeking consistency of results across methods. In addition, we would like to know whether the relationship differs depending on the main substance used.

### 3.1. Methods

#### 3.1.1. Participants

A total of 421 males with SUD were randomly selected in a compulsory drug treatment management facility in Southwest China. The 45 subjects with a correct rate of less than 0.8 for each experimental task were removed. The occurrence of low correct rates may be caused by participants’ inattentiveness or inattention. Then, 374 (90.02%) valid data were included in the analysis. 

#### 3.1.2. Study Instruments

Basic information questionnaire. A self-administered survey of demographic variables such as age, age of drug use, number of compulsory detoxifications, and type of drugs was mainly used.Depression. The instrument used was consistent with Study 1. The threshold of the BD score ≥ 8 (i.e., moderate to severe depression) was used as an assessment indicator in this study because of the need for treatment assessment for this level of depression [34]. In this study, Cronbach’s α coefficient for this scale was 0.819.Stroop color–word interference task. The Stroop color–word interference task is a test of selective attention and cognitive flexibility. It has been used in previous studies to measure individuals’ inhibitory control [35]. In the classic Stroop color–word interference task, the experimenter presents subjects with a word written in a different color and asks them to say the color of each word as quickly and as correctly as possible, regardless of the word and the meaning it represents. Three experimental conditions were included in this study: color–word agreement, color–word interference, and a control condition. In the color–word agreement condition, subjects were presented with color words written in different colors, and the name of each color word represented the same meaning as the color of the word, e.g., the Chinese word "green" was written with green paint. In the control condition, subjects were presented with neutral words in different colors. The screen background of the stimulus presentation was black. The cognitive mechanism involved in this task is called selective or directed attention because the subject must manage their attention, resist the interference of irrelevant stimuli, and inhibit or stop one response to say or do something else. The Stroop effect is usually represented by calculating the difference between the behavioral data of incongruent and congruent stimuli [36]. Color–word congruence is automatic processing, and color–word incongruence is conscious control processing.

#### 3.1.3. Experimental Design and Procedure

This study used a 2 (type of depression: high-depression group vs. low-depression group) × 2 (the main type of drug use: methamphetamine vs. heroin) between-group design with the dependent variables of the Stroop color–word interference task response time and correct rate.

After subjects arrived at the experimental area, they were first informed about the experimental procedure (subjects who did not meet the experimental requirements due to age and physical health conditions were excluded) when entering basic information and then signed an informed consent form. The subjects were grouped according to their basic information and then entered the computer room, where each computer was in a separate compartment.

#### 3.1.4. Data Analysis

The quantitative data in this study were expressed as mean and standard deviation. All data were analyzed using IBM SPSS Statistics 25.0 (Developed by IBM in the USA; https://www.ibm.com/support/pages/downloading-ibm-spss-statistics-25) (accessed on 11 January 2022). The occurrence of the Stroop effect was verified using repeated-measures ANOVA. In this study, the difference between the color–word interference condition and the control condition response was calculated to discuss inhibitory control ability as the outcome variable, and the larger the difference, the worse the inhibitory control ability. Considering that age-related variables may confound executive function (as shown in Study 1), we performed a two-factor between-groups ANOVA with the type of depression and type of first-time drug use as predictor variables, inhibition control scores as outcome variables, and age and age at first medication as control variables.

### 3.2. Results

#### 3.2.1. Demographic information of participants

In this study, the minimum age of the participants was 18, and the maximum age was 58: the mean age was 34.74 ± 7.42. The minimum age of the participants’ first drug use was 12, the maximum age was 48, and the mean age of first drug use was 24.18 ± 6.81. There were 344 participants in the methamphetamine use group and 30 in the heroin use group (detailed results are shown in Table 3).

#### 3.2.2. Stroop Effect Validation

The results showed that the difference in correct rates between the three experimental tasks was significant (*F*_(2, 375)_ = 46.069, *p* < 0.001, *η_p_^2^* = 0.197), and further post hoc tests revealed that the correct rate of the color–word agreement condition was higher than the control condition (*p* < 0.05) and the color–word interference condition (*p* < 0.05). The difference in response time between the three experimental tasks was significant (*F*_(2, 375)_ = 296.251, *p* < 0.001, *η_p_^2^* = 0.612), and further post hoc tests revealed that the color–word agreement condition and response time were lower than the control condition response time and the color–word interference condition response time (*p* < 0.001). This result suggests that a Stroop effect occurred.

#### 3.2.3. ANOVA Results for Inhibitory Control Scores

The results are shown in Figure 1. The main effect of depression was significant (M_low-depression_ = 34.21, SD_low-depression_ = 65.81, M_High-depression_ = 46.46, SD_High-depression_ = 73.52; *F*_(1, 368)_ = 5.435, *p* = 0.020, *η_p_^2^* = 0.015), the main effect of the type of first-time drug use was not significant (M_low-depression_ = 30.44, SD_low-depression_ = 69.76, M_High-depression_ = 80.62, SD_High-depression_ = 72.05; *F*_(1, 368)_ = 0.751, *p* = 0.387, *η_p_^2^* = 0.002), and the interaction effect between the two was not significant (*F*_(1, 368)_ = 1.995, *p* = 0.159, *η_p_^2^* = 0.005).

### 3.3. Discussion of Study 2

Study 2 used an experimental paradigm and found results largely consistent with previous ones: cognitive inhibition was significantly lower in the high-depression group than in the low-depression group. In addition to this, the findings showed a non-significant effect of the type of substance use (methamphetamine and heroin) in the relationship between depression and cognitive inhibition. This suggests that the effect of depression on cognitive suppression in addicted patients participating in mandatory drug treatment for the first time is not influenced by the type of drug use.

## 4. General Discussion

We used a cross-sectional survey to find that depression negatively predicted cognitive inhibition in patients participating in compulsory drug treatment for the first time. Then, we used the Stroop paradigm to find that cognitive inhibition was worse in high-depression patients compared to low-depression patients. Both studies provided some evidence that depression can predict decreased inhibition in SUD populations. This result suggests a special angle of discussion to address factors associated with impaired inhibitory function in drug addicts and provides new enlightenment for the current field of working memory training for substance addiction.

The effect of depression on cognitive inhibition has been demonstrated in the general population. For example, Colich et al. found that depressed individuals show abnormalities in the prefrontal cortex during inhibition tasks in response to negative stimuli compared to normal individuals, i.e., there is an impairment in inhibitory control [37]. According to cognitive theory, depressed individuals develop a negative cognitive schema based on early traumatic experiences that leads to depressive beliefs wherein they view themselves, the world around them, and their future pessimistically [38]. The schema is activated with a diminished ability to inhibit negative automatic thinking when processing subsequent information, creating a negative cognitive bias in attention, assessment, and other areas [25,39]. A meta-analysis of the Stroop mood study showed that depressed patients were subject to stronger negative stimulus interference effects compared to healthy controls [40,41] and that the interference effects were greater, with higher severity of depressive symptoms [40]. It was confirmed that it is more difficult to inhibit negative stimuli in depressed patients. This effect should also exist in the addicted population.

The results of both Study 1 and Study 2 support the hypothesis that depression in people with SUD may exacerbate the degree of impairment in their inhibitory control functions. Specifically, inhibitory control scores were significantly lower in the highly depressed group than in the non-depressed group, conditional on the control age and age of first drug use. The dual-systems model suggests [42] that impulsive behaviors such as drug use arise primarily due to the activation of impulsive systems, which are several automatic emotional responses and behavioral association tendencies derived from long-term experience and learning, activated by specific social situations. This process is automated and requires little involvement of cognitive resources [43]. The inhibition system, on the other hand, functions as a primary reflection of the individual’s self-control or strength of willpower and includes deliberate evaluation and inhibition criteria [44]. This system is relatively slow due to the need for individuals to consciously employ symbol-based representational systems and operating systems. Control systems depend more on control resources and are more proactive [45,46].

However, according to the depression susceptibility dual-processing theory, depressed individuals or those with high scores on depression scales engage in negative automatic processing and produce negative automatic ruminative thinking, which diminishes their cognitive resources and impairs the resources allocated to other tasks [26]. Substance use by addicted individuals can alleviate or eliminate states of discomfort or aversion within the body, a behavior that may be reinforced through negative reinforcement processes [47]. Individuals continue to use drugs to alleviate uncomfortable states such as those associated with negative emotional states, tension, arousal, craving, or withdrawal, and these states persist throughout the individual’s withdrawal period [48]. These uncomfortable states also serve as sensitive cues which the addict preferentially processes in response. Processing of these cues also diminishes their cognitive resources and, in turn, impairs resources allocated to other tasks. Thus, in the Stroop task, individuals need to mobilize brain resources to cope with inconsistent information. Insufficient cognitive resources at this point can limit the operation of the control system and make it difficult for individuals to inhibit dominant responses [48]. Thus, subjects in this category show a slower response to color stimuli when the word meaning is inconsistent with the color.

Furthermore, the present study examined the role of the type of substance use (methamphetamine and heroin) in the relationship between depression and cognitive inhibition, and no significant interaction was found. This result shows, to some extent, that the different pharmacological effects of the two types of addiction sources do not affect the relationship between depression and cognitive inhibition.

Once severe drug addictive behaviors have developed, there is no definitive treatment available, except for some that can be treated with drug substitution. However, high relapse rates can occur after a drug addict is released from mandatory isolation. For this reason, various disciplines have explored treatment modalities from different perspectives. Neurobiological and psychological studies of drug addiction find that people with SUD are susceptible to impairments in inhibition function [49], which is relevant to the treatment outcome of substance use disorders [50]. Most current research suggests that working memory or executive function training can improve cognitive performance in addicts, but the effects of training and the transfer it produces are inconsistent. Many researchers have attributed this to the site and extent of brain damage and the resulting different cognitive deficits caused by different types of drugs in addicts [51,52], and some have also focused on the differences between pathological states that can be caused by individuals of different genders [53]. However, less attention has been paid to the impact of comorbidities associated with substance addiction, yet people with SUD often suffer from other disorders, creating a state of comorbidity [54]. This study found a strong association between impaired cognitive inhibition and depressive symptoms in people with SUD, suggesting that the impact of patients’ psychiatric comorbidities should be attended to during working memory training treatment for substance addicts. Additionally, this result has important implications for standard screening in the early stages of substance addiction treatment to define the treatment process and treat patients with comorbidities such as addiction and depression. 

Although we provided a credible predictive result through larger samples and two methods, we still cannot explain causality. In the future, animal experiments can be used to further clarify the causal relationship between depression and the impairment of inhibition function in addicts. We compared the effect of drug type on cognitive depression in Study 2, but the small number of people in the heroin group relative to the methamphetamine group due to sampling limitations makes this result potentially less credible. The number of people in the heroin group could be increased in the future to obtain more reliable results. Since cognitive suppression may depend on the time of discontinuation, we did not consider the time of discontinuation of participants and could focus on this variable in the future. The present study also only discussed the difference in cognitive inhibition between addiction in depressed patients and non-depressed patients and did not discuss the underlying mechanism of depression-induced cognitive inhibition impairment, which needs to be further discussed by incorporating physiological and brain imaging data in the future. Nevertheless, our work contributed to the literature on discovering the negative relationship between depression and cognitive inhibition among addicts. It enlightened the researchers to select more proper solutions for helping the addicts to overcome their mental health problems. Considering that depression would induce impairment of cognitive inhibition, the cognitive and behavioral therapies that need cognitive inhibition might not be the way to go for people with comorbidity of addiction and depression. Other therapies might be more useful, such as motivational interviewing or contingency management.

## 5. Conclusions

The two studies in this article showed that cognitive inhibition is weaker in people with high-depression addiction than in those with low depression. This result reminds us that in the treatment of substance addiction with working memory training, we need to pay attention to the role of comorbidities such as depression.

## Figures and Tables

**Figure 1 healthcare-11-00070-f001:**
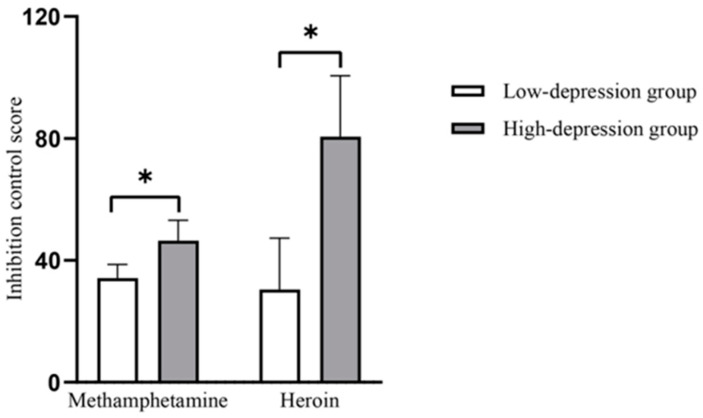
Inhibitory control scores in the between-subject groups. Note: * *p* < 0.05.

**Table 1 healthcare-11-00070-t001:** Results of descriptive statistics and correlation analysis of depression and impulse inhibition.

Variable	1	2	3	4
1. Age	1			
2. FDG	0.687 ***	1		
3. Depression	−0.005	−0.023	1	
4. SII	0.124 ***	0.117 ***	−0.218 ***	1
M	34.47	24.33	7.05	3.204
SD	7.360	6.946	5.240	0.750

Note: *** *p* < 0.001, same below. FDG = age of first drug use; SII = stimulation-induced inhibition.

**Table 2 healthcare-11-00070-t002:** Results of regression analysis of depression and impulse inhibition.

Predictor Variable	Outcome Variable: SII				
	*b*	SE	*t*	*p*	95% CI
Constant	3.008	0.151	19.975	<0.001	(2.712, 3.303)
Age	0.008	0.006	1.434	0.152	(−0.003, 0.019)
FDG	0.006	0.006	0.942	0.346	(−0.006, 0.017)
Depression	−0.031	0.006	−5.281	<0.001	(−0.043, −0.020)
Joint explanatory power	*R*^2^ = 0.062***				
Overall Significance	*F*_(3,567)_ = 12.462***				

Note: *** *p* < 0.001; b = non-standardized regression coefficients; FDG = age of first drug use; SII = stimulation-induced inhibition.

**Table 3 healthcare-11-00070-t003:** Demographic information of participants (*n* = 374).

Type of First-Time Drug Use	Type of Depression	*n* (%)	Age		FDG	
			M	SD	M	SD
Methamphetamine	Low depression	223 (59.63)	33.18	6.897	24.03	7.092
	High depression	121 (32.35)	34.27	6.613	24.74	6.621
Heroin	Low depression	17 (4.54)	42.94	7.949	23.35	5.314
	High depression	13 (3.48)	45.69	8.087	23.23	5.732

Note: *n* = number of participants; FDG = age of first-time drug use.

## Data Availability

The data that support the findings of this study are available from the corresponding author, upon reasonable request.

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
