# Peer review of "Relationship between Depression and Cognitive Inhibition in Men with Heroin or Methamphetamine Use Disorder in First-Time Mandatory Detoxification"

_healthcare, 2022, doi:10.3390/healthcare11010070_

Round 1
Reviewer 1 Report
Dear authors,
I've read this study with interest. Overall it is well-written and comprehensive. I have some major and minor suggestions in order to improve the quality and suitable for publication.
First of all, I have never seen another section of methods and results following a first section of methods and results. This should either be split into two seperate papers, but as it is I strongly suggest you move the whole Stroop methodology to the methods section and number through accordingly (2.7, 2.8..). Because now it is very confusing and not comprehensive for the reader. Also group your results in this way, the Stroop results just follow the overall regression results of impulse impairment and depression. You might want to keep subsections of "study 1" and "study 2" in your results. Just state that you continued with another sample for the Stroop results (either in methods or in results).
Also, do not use references or discussions in your results section! This should all be reserved for your final discussion. So replace most of 2.3 to discussion and the the same for 3.3.
In the general discussion you start of with a sentence on 283 "This difference provides some evidence that depression also predicts diminished inhibitory function in people with SUD [22, 23]." NEVER start your discussion with referring to other evidence! Your first paragraph should summarize the main findings in your own study (or two studies) and what this might imply in general, practical consequences or meaning.
Only then begin with paragraphs with literature that either contradicts or supports your findings. This also makes a lot easier read for the audience.
In line 356 you actually diminish your results, by stating that this study "simply" explored. This does not do your interesting results justice! Be more proud of what you've done and the number of participants that were enrolled. You found interesting results that have important implications for treating patients with addiction and a co-morbidity like depression. So re-formulate the last paragraph and summarize the findings in the context of this. For example, you might indicate that CBT (cognitive therapy) might not be the way to go for people with co-morbidity of addiction and depression, but other therapies might be more useful, like motivational interviewing or contingency management.
Author Response
Manuscript ID: healthcare-2064509
Manuscript Title: Relationship between depression and cognitive inhibition in men with heroin or methamphetamine use disorder in first-time mandatory detoxification
Dear Editors and Reviewers,
On behalf of my co-authors, we sincerely thanks for giving us an opportunity to revise our manuscript. We appreciate the Editor and Reviewers very much for the valuable feedback for improving the quality of our manuscript. The manuscript has been carefully revised and proofread according the comments.
Response to Reviewer 1 Comments
Reviewer 1
Dear authors,
I've read this study with interest. Overall it is well-written and comprehensive. I have some major and minor suggestions in order to improve the quality and suitable for publication.
Q: First of all, I have never seen another section of methods and results following a first section of methods and results. This should either be split into two seperate papers, but as it is I strongly suggest you move the whole Stroop methodology to the methods section and number through accordingly (2.7, 2.8..). Because now it is very confusing and not comprehensive for the reader. Also group your results in this way, the Stroop results just follow the overall regression results of impulse impairment and depression. You might want to keep subsections of "study 1" and "study 2" in your results. Just state that you continued with another sample for the Stroop results (either in methods or in results).
A: We appreciate this suggestion. Given the whole structure, we think it is better to leave it as it is now. This paper is a paper with two studies. Thus, there are two parts of method and results. Maybe this structure is not very common seen in the reviewer’s discipline background. However, it is quite often seen in some disciplines, such as Psychology. Also, we can see this kind of structure in the published papers on Healthcare, for example,
Mora-Pelegrín, M.; Montes-Berges, B.; Aranda, M.; Vázquez, M.A.; Armenteros-Martínez, E. The Empathic Capacity and the Ability to Regulate It: Construction and Validation of the Empathy Management Scale (EMS). Healthcare 2021, 9, 587. https://doi.org/10.3390/healthcare9050587
In our paper, we conducted two studies to test the relationship between depression and cognitive inhibition. The studied population is the men with heroin or methamphetamine use disorder in first-time mandatory detoxification. We predicted that depression might induce the decrease of cognitive inhibition ability. In Study 1, we tested this prediction with a cross-sectional survey. However, the cognitive inhibition might not be well measured just with a self-reported scale. To measure the cognitive inhibition more specifically, we conducted Study 2 and used Stroop task to detect cognitive inhibition. To summarize, these two studies are fundamentally two different and related studies. We think it is better to leave it as it is now so that the whole logic could be presented.
Thanks again. We hope we have made it clear this time.
Q: Also, do not use references or discussions in your results section! This should all be reserved for your final discussion. So replace most of 2.3 to discussion and the the same for 3.3.
A: Thanks for this professional suggestion. We agree that we should not use references or discussions in the results section. We have removed the references. Regarding the sections of 2.3 and 3.3, we have revised the section titles to be “Discussion of Study 1” and “Discussion of Study 2”. We hope this would be clearer given that we hope to remain the two studies structure.
Q: In the general discussion you start of with a sentence on 283 "This difference provides some evidence that depression also predicts diminished inhibitory function in people with SUD [22, 23]." NEVER start your discussion with referring to other evidence! Your first paragraph should summarize the main findings in your own study (or two studies) and what this might imply in general, practical consequences or meaning. Only then begin with paragraphs with literature that either contradicts or supports your findings. This also makes a lot easier read for the audience.
A: Thanks for the insightful suggestion. We agree and learned the tips for writing the general discussion. In the revised version of the general discussion, we begain with a paragraph summarizing the results of our two studies. Then, we explained and discussed our results by comparisons with previous studies. In this way, we have discussed how previous theories could explain our results and support our findings, and also discussed the limitations of our studies and respective future directions.
Q: In line 356 you actually diminish your results, by stating that this study "simply" explored. This does not do your interesting results justice! Be more proud of what you've done and the number of participants that were enrolled. You found interesting results that have important implications for treating patients with addiction and a co-morbidity like depression. So re-formulate the last paragraph and summarize the findings in the context of this. For example, you might indicate that CBT (cognitive therapy) might not be the way to go for people with co-morbidity of addiction and depression, but other therapies might be more useful, like motivational interviewing or contingency management.
A: We really appreciate your high evaluations on our paper. We agree that we contributed a lot with large number of participants and obtained interesting results. However, we think we should rigorously discuss the limitations of our work. Thus, we have given a short paragraph to report what we think could promote the study in the future. To further emphasize our contribution, we have added a sentence in the last paragraph, i.e.,
Nevertheless, our work contributed to the literature on discovering the negative relation-ship between depression and cognitive inhibition among addicts. It enlightened the re-searchers to select the more proper solutions for helping the addicts out from the mental health problem. Given that the depression would induce impairment of cognitive inhibi-tion, the cognitive and behavioral therapies that need cognitive inhibition might not be the way to go for people with co-morbidity of addiction and depression. Other therapies might be more useful, like motivational interviewing or contingency management.
Special thanks to Reviewer#1 for the excellent suggestions again and sincerely hope that the correction will meet with approval.

Reviewer 2 Report
Authors studied the correlation of depression and cognitive inhibition in heroin and methamphetamine users in first time mandatory detoxification. Please see the below comments.
1. In the 3.1.1 participants section, authors described the number of patients in non-depressed and depressed group. But again, in the 3.1.2 Study instrument section, it was mentioned that Bd score >= 8 (moderate to severe depression) score was used as an assessment indicator. Please clarify.
2. Authors did not present the stroop effect scores and the mean and SD scores for control, lower and higher depression groups. Also, please add in the methods section about the ranges of the stroop effect scores.
3. There is a significant difference in the number of participants in the methamphetamine and heroin group. How did the authors come to a conclusion that there is not a significant effect between methamphetamine and heroin group, given the significant difference between the two groups. Please clarify.
4. Results in the abstract also mentioned “ Results showed that the depressed group has weaker inhibition performances in the stroop color-word interference task compared to non depressed group. This study shows that cognitive inhibition is weaker in people with high depression addiction than in those with low depression.” Please clarify the use of both non-depressed and low depression as methods suggested the selection of moderate to severe depression patients. Please clarify.
5. Is the withdrawal time of participants considered as the cognitive inhibition can be dependent on drug withdrawal time? Please clarify.
6. In the 2.2.1 section table 1, *** is shown to explain significance for all the results whereas in the description above table-1, it was mentioned that “Correlation analysis showed that depression was significantly negatively associated with impulse inhibition (p< 0.01). Please clarify.
7. In the 2.3 results section, results from cognitive inhibition were explained whereas study 1 experiments described impulsive inhibition.
8. There are two sections of Experimental design and Procedure, 3.1.3 and 3.1.4. Suggest to combine to 1 section.
9. Study 2 has a discussion section 3.3. Suggest to move this to 4. Discussion section.
10. Suggest to rewrite first three lines in the introduction section and add references.
11. Please remove “571 males with SUD” from the first line in the 2.1.3 Data processing and common method deviation test section.
12. R-square seems to be low as 0.062 as mentioned in table-2. Please clarify.
13. From Table-1, mean score of depression is 7.05 and SD is 5.240. Please specify the number of patients in high depression and low depression group as in study-2.
Author Response
Manuscript ID: healthcare-2064509
Manuscript Title: Relationship between depression and cognitive inhibition in men with heroin or methamphetamine use disorder in first-time mandatory detoxification
Dear Editors and Reviewers,
On behalf of my co-authors, we sincerely thanks for giving us an opportunity to revise our manuscript. We appreciate the Editor and Reviewers very much for the valuable feedback for improving the quality of our manuscript. The manuscript has been carefully revised and proofread according the comments.
Response to Reviewer 2 Comments
Reviewer 2
Authors studied the correlation of depression and cognitive inhibition in heroin and methamphetamine users in first time mandatory detoxification. Please see the below comments.
Q1. In the 3.1.1 participants section, authors described the number of patients in non-depressed and depressed group. But again, in the 3.1.2 Study instrument section, it was mentioned that Bd score >= 8 (moderate to severe depression) score was used as an assessment indicator. Please clarify.
A: Thanks for this comment. To ensure logical consistency and avoid ambiguity, we removed the introduction to grouping in the 3.1.1 participants section and added a new section (in the 3.2.1. Demographic information of participants) describing the demographics of the participants in the 3.2 results section.
Q2. Authors did not present the stroop effect scores and the mean and SD scores for control, lower and higher depression groups. Also, please add in the methods section about the ranges of the stroop effect scores.
A: Thank you for this important comment. We added the mean and standard deviation of the Stroop scores for each group in the 3.2.3. ANOVA results for inhibitory control scores section.
Q3. There is a significant difference in the number of participants in the methamphetamine and heroin group. How did the authors come to a conclusion that there is not a significant effect between methamphetamine and heroin group, given the significant difference between the two groups. Please clarify.
A: Thanks for your suggestion. Due to sampling limitations, we have always been unable to balance the number of two people, but heroin and methamphetamine are two different drugs, we hope to see their role in our results. Therefore, our results are more cautious. And according to the reviewer 's reminder, we also describe this situation in the limited section.
Q4. Results in the abstract also mentioned “ Results showed that the depressed group has weaker inhibition performances in the stroop color-word interference task compared to non depressed group. This study shows that cognitive inhibition is weaker in people with high depression addiction than in those with low depression.” Please clarify the use of both non-depressed and low depression as methods suggested the selection of moderate to severe depression patients. Please clarify.
A: Thanks for pointing this out. We are sorry about this error. This is a mistake we made when writing and we have corrected it to “high-depressed group” and “low- depressed group”.
Q5. Is the withdrawal time of participants considered as the cognitive inhibition can be dependent on drug withdrawal time? Please clarify.
A: Thank you for your reminding, but we didn 't take this into consideration. One reason is that we are not particularly aware of the specific measures of compulsory detoxification. In addition, after your reminder, we checked the time of participants participating in compulsory detoxification ( from 1 month to 24 months ), and the distribution was random, which could balance this result to some extent.
Q6. In the 2.2.1 section table 1, *** is shown to explain significance for all the results whereas in the description above table-1, it was mentioned that “Correlation analysis showed that depression was significantly negatively associated with impulse inhibition (p< 0.01). Please clarify.
A: Thanks for your comment. We correct p < 0.01 to p < 0.001 and optimize the annotation part of the whole table.
Q7. In the 2.3 results section, results from cognitive inhibition were explained whereas study 1 experiments described impulsive inhibition.
A: Thanks for your query. This is a mistake we made when writing and we have corrected it to stimulation-induced inhibition
Q8. There are two sections of Experimental design and Procedure, 3.1.3 and 3.1.4. Suggest to combine to 1 section.
A: We are sorry for that. This is a mistake we made when writing and 3.1.4. should be the data analysis section. We have corrected it.
Q9. Study 2 has a discussion section 3.3. Suggest to move this to 4. Discussion section.
A: Thank you for the suggestion. As Section 3.3 is a simple discussion of the results of study 2, and section 4 is a general discussion of the results of the whole paper, so it may be better to put section 3.3 in part 3.
Q10. Suggest to rewrite the first three lines in the introduction section and adding references.
A: Thank you for this comment. We have rewritten the first three lines in the introduction section and added references
Q11. Please remove “571 males with SUD” from the first line in the 2.1.3 Data processing and common method deviation test section.
A: We are sorry. This is a mistake we made when writing and we removed it.
Q12. R-square seems to be low as 0.062 as mentioned in table-2. Please clarify.
A: Thanks for your comment. Although the R-square looks low, it is statistically significant (F = 12.462, p < 0.001). From the perspective of statistics, R-square would be smaller with increase of sample size. Given that we sampled hundreds of participants, this might partly explain why R-square is as low as 0.062. Thanks again.
Q13. From Table-1, mean score of depression is 7.05 and SD is 5.240. Please specify the number of patients in high depression and low depression group as in study-2.
A: Thanks. We added this result in section 3.2.1.
Special thanks to Reviewer#2 for the excellent suggestions again and sincerely hope that the correction will meet with approval.

Round 2
Reviewer 1 Report
Most criticisms are met now, except for the split into two sections, but I guess that this unorthodox way of presenting results is more common in this journal. It doesn't affect the quality too much
Reviewer 2 Report
Thank You for providing the responses.